# Antigen-Specific IFN-γ/IL-17-Co-Producing CD4^+^ T-Cells are the Determinants for Protective Efficacy of Tuberculosis Subunit Vaccine

**DOI:** 10.3390/vaccines8020300

**Published:** 2020-06-11

**Authors:** Han-Gyu Choi, Kee Woong Kwon, Seunga Choi, Yong Woo Back, Hye-Soo Park, Soon Myung Kang, Eunsol Choi, Sung Jae Shin, Hwa-Jung Kim

**Affiliations:** 1Department of Microbiology, and Medical Science, College of Medicine, Chungnam National University, Daejeon 35015, Korea; ekdrms20000@gmail.com (H.-G.C.); seungachoi@cnu.ac.kr (S.C.); lenpk@nate.com (Y.W.B.); 01027192188@hanmail.net (H.-S.P.); 2Department of Microbiology, Institute for Immunology and Immunological Diseases, Brain Korea 21 PLUS Project for Medical Science, Yonsei University College of Medicine, Seoul 03722, Korea; KKEEWEE@yuhs.ac (K.W.K.); wowwow08@yuhs.ac (S.M.K.); eunsolnoel@yuhs.ac (E.C.)

**Keywords:** *Mycobacterium tuberculosis*, BCG-prime boost, IFN-γ/IL-17, multifunctional T cells, phagosome maturation

## Abstract

The antigen-specific Th17 responses in the lungs for improved immunity against *Mycobacterium tuberculosis* (Mtb) infection are incompletely understood. Tuberculosis (TB) vaccine candidate HSP90-ESAT-6 (E6), given as a Bacillus Calmette-Guérin (BCG)-prime boost regimen, confers superior long-term protection against the hypervirulent Mtb HN878 infection, compared to BCG or BCG-E6. Taking advantage of protective efficacy lead-out, we found that ESAT-6-specific multifunctional CD4^+^IFN-γ^+^IL-17^+^ T-cells optimally correlated with protection level against Mtb infection both pre-and post-challenge. Macrophages treated with the supernatant of re-stimulated lung cells from HSP90-E6-immunised mice significantly restricted Mtb growth, and this phenomenon was abrogated by neutralising anti-IFN-γ and/or anti-IL-17 antibodies. We identified a previously unrecognised role for IFN-γ/IL-17 synergism in linking anti-mycobacterial phagosomal activity to enhance host control against Mtb infection. The implications of our findings highlight the fundamental rationale for why and how Th17 responses are essential in the control of Mtb, and for the development of novel anti-TB subunit vaccines.

## 1. Introduction

Tuberculosis (TB), an infectious disease caused by *Mycobacterium tuberculosis* (Mtb), is associated with high morbidity and mortality, thus posing a global public health problem. In 2017, TB ranked as one of the top ten causes of death, with an estimated 10 million new cases and 1.6 million deaths [1]. In addition, approximately 1.7 billion people, 23% of the global population, are estimated to have latent TB infection and to be at risk of developing active TB during their lifetime. In addition, the emergence of Mtb strains resistant to TB drugs poses a major growing burden of hard-to-treat infections [2].

*Mycobacterium bovis* Bacillus Calmette-Guérin (BCG) currently is the only licensed prophylactic vaccine; however, it provides insufficient protection against TB, and thus, novel effective vaccines are urgently needed [3]. Various types of adjuvants, antigen (Ag) targets, and vaccine platforms have been developed in an aim to improve Mtb vaccines. These efforts have yielded various results, with some producing positive outcomes in clinical trials. Heterologous prime-boost regimens involving priming with BCG, followed by an adjuvant boost, are a promising vaccination strategy against TB [4], and have a proven high level of efficacy.

Clinical efficacy trials of three TB multi-antigenic subunit vaccines (H4:IC31, M72/AS01_E_, and ID93) conducted in 2018 yielded encouraging results, and helped to advance experimental design strategies in the field of TB vaccine development [5,6,7]. All three subunit vaccine candidates are multi-antigen, single-fusion protein vaccines formulated with their own unique adjuvant, and have been evaluated in BCG-vaccinated healthy populations in TB-endemic regions, mainly in South-African countries. The candidates effectively boosted a BCG-induced immune response and provided long-term protection and induced persistent Th1-biased multifunctional CD4^+^ T-cell responses in preclinical TB models [8,9].

We previously demonstrated that a subunit vaccine consisting of the ESAT-6 Mtb antigen fused with HSP90 (hereafter referred to as HSP90-E6) formulated with MPL/dimethyldioctadecyl- ammonium (DDA) as an adjuvant confers high-level, robust protection against the hypervirulent Beijing strain, HN878 [10]. The improved protection provided by this vaccine was characterised by durable, robust pulmonary Th1-polarised multifunctional CD4^+^ T-cell immune responses in the lungs in comparison with BCG or ESAT-6 alone in a standard mouse model [10]. These findings suggested the potential usability of this vaccine candidate.

Similar to MPL, CIA05 is a TLR4 agonist purified from an *Escherichia coli* strain that expresses lipopolysaccharides with short carbohydrate chains and detoxified by alkaline hydrolysis [11]. CIA05 stimulates the secretion of various cytokines and chemokines from human monocytes and mouse bone-marrow dendritic cells (DCs), and the immuno-stimulatory activity of CIA05 is higher than that of MPL [11]. Therefore, in the current study, we tested the efficacy of HSP90-E6 TB vaccine with CIA05 instead of MPL adjuvant.

For a vaccine to induce protection against TB, antigen-specific T-cells should be rapidly recruited to the lungs and activate the infected phagocytes [12,13]. While CD4^+^IFN-γ^+^ T-cells are generally thought to be essential for Mtb control [14,15], IFN-γ production does not correlate with protection against TB [16,17,18]. Moreover, recent data suggest that CD4^+^ T-cells producing multiple cytokines, including IFN-γ, TNF-α, and IL-2, are associated with protection against TB [19,20,21,22], suggesting that multifunctional CD4^+^ T-cells are important in Mtb control. Further, IL-17 and Th17 responses have been found to be important for protective immunity against TB [13,23,24,25,26,27,28,29]. CD4^+^IL-17^+^ T-cells play a particularly crucial role in vaccine-mediated immunity [13,30] by promoting a prompt recruitment of CD4^+^IFN-γ ^+^ T-cells to the lungs, leading to early control of Mtb replication in Mtb-infected mice [13]. However, the function of IL-17 in the protection against Mtb and, in particular, in synergism with IFN-γ in Mtb-infected macrophages, remains unclear. Our previous study investigating cytokine profiles from DCs activated by E6-HSP90 treatment and co-cultured with naïve CD4^+^ T-cells suggested a possible involvement of the Th17 response [10]; however, whether the protective mechanism of this vaccine candidate is associated with its Th17-inducing capacity remained to be clarified.

In the current study, a heterologous prime-boost regimen with HSP90-E6/CIA05 following BCG vaccination was used to evaluate putative correlations between protective efficacy and CD4^+^ T-cell subset phenotypes. In addition, we investigated whether a Th17 response is required for the optimal efficacy of this vaccine candidate, as well as the underlying mechanism.

## 2. Materials and Methods

### 2.1. Ethics Statement

All animal studies were performed in accordance with Korean Food and Drug Administration (KFDA) guidelines. The experimental protocols used in this study were reviewed and approved by the Ethics Committee and Institutional Animal Care and Use Committee (Permit Number: 2014-0197-3) of the Laboratory Animal Research Center at Yonsei University College of Medicine (Seoul, Korea) and IACUC (CNU-00284) of animal care at Chungnam National University (Daejeon, Korea).

### 2.2. Mice

Specific pathogen-free female C57BL/6J mice (6–7 weeks old) were purchased from Japan SLC, Inc. (Shizuoka, Japan), and maintained under barrier conditions in the ABSL-3 facility at the Yonsei University College of Medicine with constant temperature (24 °C ± 1 °C) and humidity (50% ± 5%). The animals were fed a sterile commercial mouse diet with ad libitum access to water under standardized light-controlled conditions (12-h light and 12-h dark periods). The mice were monitored daily, and none of the mice showed any clinical symptoms or illness during this experiment.

### 2.3. Preparations of Mtb Strains and Antigens

Mtb HN878 was obtained from the strain collections of the International Tuberculosis Research Center (ITRC, Changwon, Gyeongsangnam-do, South Korea). BCG (Pasteur strain 1173P2) was kindly provided by Dr. Brosch, from the Pasteur Institute (Paris, France). All mycobacteria used in this study were prepared as described previously [31].

PPD was kindly provided by Dr. Michael Brennan at Aeras (Rockville, MD, USA). To produce a recombinant HSP90-E6 protein, the corresponding gene was amplified by PCR using Mtb H37Rv ATCC27294 genomic DNA as a template and the following primers: HSP90 forward, 5′- CATATGAACGCCCATGTCGAGCAGTTG-3′, and reverse, 5′-GAATTCGGCAAGGTACGCGCG AGACGTTC-3′; ESAT-6 forward, 5′-AAGCTTATGACAGAGCAGCAGTGGAAT-3′, and reverse, 5′-CTCGAGTGCGAACATCCCAGTGACGTT-3′. The PCR product of HSP90 was digested with *NdeI* and *EcoRI*, and ESAT-6 was cut with *HindIII* and *XhoI*. The products were inserted into the pET22b (+) vector (Novagen, Madison, WI, USA), and the resultant plasmids were sequenced. The recombinant plasmids were transfected into *E. coli* BL21 cells by heat-shock for 1 min at 42 °C. To produce a recombinant fusion protein, the PCR products of HSP90 were inserted into previously produced ESAT-6-containing pET22b (+) vector. The recombinant protein was prepared as previously described [10].

### 2.4. Immunisation and Mtb Infection in Mice

C57BL/6 mice (*n* = 12/group) were injected subcutaneously with a single dose of 2 × 10^5^ colony forming units (CFU) of BCG Pasteur 1173P2 (prime), and were immunised 3 months later, through 3 intramuscular injections administered 3 weeks apart (boosts). Each immunisation contained 2 μg of ESAT-6 and HSP90-E6 with 2 μg of CIA05 formulated in 250 μg of dimethyldioctadecylammonium (DDA) liposomes. The control group was immunised with CIA05/DDA only. Immunogenicity was analysed in spleen and lung cells 4 weeks after the last immunisation. Afterwards, immunised mice (CIA05/DDA, BCG, BCG+ESAT-6/CIA05/DDA, and BCG+HSP90-E6/CIA05/DDA) were challenged by aerosol exposure with Mtb HN878 strain, as previously described [32]. Briefly, mice were exposed to HN878 strain in the calibrated inhalation chamber of an airborne infection apparatus for 60 min, delivering a predetermined dose (Glas-Col, Terre Haute, IN, USA); approximately 200 viable bacteria were delivered.

### 2.5. Intracellular Cytokine Staining

After 10 wk Mtb HN878 challenge, the mice were euthanized by CO_2_ asphyxiation, and single cell suspensions (1 × 10^6^ cells) from immunised and infected mice were stimulated with PPD (2 μg/mL) or ESAT-6 (2 μg/mL) at 37 °C for 12 h in the presence of both GolgiPlug and GolgiStop (BD Biosciences, San Jose, CA, USA). PBS-washed cells were blocked with anti-CD16/32 (BD Biosciences) at 4 °C for 20 min. After that, cells were surface stained with Brilliant Violet (BV) 605-conjugated anti-CD90.2, peridinin chlorophyll (PerCP)-Cy5.5-conjugated anti-CD4 (BD Biosciences), allophycocyanin (APC)-Cy7-conjugated anti-CD8 (Biolegend, San Diego, CA, USA), and V450-conjugated anti-CD44 antibodies at 4 °C for 30 min, and were washed three times with PBS. These cells were fixed and permeabilised with the Cytofix/Cytoperm kit (BD Biosciences) at 4 °C for 30 min. Then, cells were washed three times with Perm/Wash (BD Biosciences) and stained intracellularly with PE-conjugated anti-IFN-γ, PE-Cy7-conjugated anti-IL-2, APC-conjugated anti-TNF-α and Alexa488-conjugated anti-IL-17 (T-bet, GATA-3 and RORγt) (BD Biosciences) at 4 °C for 30 min. After being washed three times with Perm/Wash, cells were fixed using IC Fixation buffer (eBioscience). Then, PBS-resuspended cells were analysed on a CytoFLEX S Flow cytometer (Beckman Coulter, Indianapolis, IN, USA), using the commercially available software program FlowJo (Treestar, Inc., San Carlos, CA).

### 2.6. Bacterial Counts and Histopathological Analysis

Adherent bone marrow-derived macrophages (BMDMs) (2 × 10^5^ cells/well) were washed twice in PBS and infected, in triplicate with Mtb (2 × 10^5^ bacilli/well). Tubercle bacilli and macrophages were incubated for 4 hr. Then, the infected BMDMs were treated with amikacin (200 μg/mL) for 2 hr. After 2 hr, monolayers were washed to remove extracellular bacilli, and this time point was considered as day 0. Two different types of measurement of CFU in infected macrophages were used. First, the infected BMDMs were preincubated with neutralizing antibody against IFN-γR1 (R&D Systems, Minneapolis, MN, USA) (200 ng/mL), IL-17RA (R&D Systems) (200 ng/mL), or IFN-γR1/IL-17RA (200 ng/mL) for 2 hr, and then a previously prepared culture supernatants—antigen-activated DCs co-cultured with CD4^+^ T-cells at a DC:T-cell ratio of 1:10 for 3 days—was added to each well, and the plate was incubated for 3 days. The following DC-activating antigens were used: PPD, ESAT-6, and HSP90-E6 (2 μg/mL). Second, infected BMDMs were treated with recombinant mIFN-γ (R&D Systems) (10 ng/mL), recombinant mIL-17 (R&D Systems) (10 ng/mL), or recombinant mIFN-γ/mIL-17 (1 ng/mL) for 3 days. The number of ingested and internalised Mtb by the BMDMs was calculated by lysing the infected cells from one of the wells in distilled water. The Tubercle bacilli counts in the inoculum were then checked by serial dilution and plating on 7H10 agar with 10% Middlebrook OADC supplement (Difco, Detroit, MI). The plates were incubated at 37 °C for 3 weeks. Afterwards, the plates were taken out, and colony forming units (CFUs) were calculated based on the number of colonies of Mtb.

Ten weeks after Mtb HN878 infection, the lungs and spleens dissected from the infected mice were homogenised. The number of viable bacteria was determined by plating serial dilutions of the organ (left lung or half spleen) homogenates onto Middlebrook 7H11 agar (Difco Laboratories, Detroit, MI, USA) supplemented with 10% OADC (Difco Laboratories), amphotericin B (Sigma-Aldrich, St. Louis, MO, USA), and 2 μg/mL 2-thiophenecarboxylic acid hydrazide (Sigma-Aldrich). Colonies were counted after 4 weeks of incubation at 37 °C. For the histopathological analysis, the superior lobes of the right lung were vertically or horizontally sectioned, and stained with Haemotoxylin and Eosin (H&E). For assessment of severity of inflammation, representative horizontal sections were used. The level of inflammation and the size of lesions in the lungs was evaluated using the ImageJ software (National Institutes of Health, Bethesda, MD), as described previously [32].

### 2.7. Quantification of Cytokines

A sandwich enzyme-linked immunosorbent assay (ELISA) was used for detecting IL-1β, TNF-α, IFN-γ, IL-4, IL-2, IL-12p70, IL-12p19, IL-17, and IL-10 in culture supernatants, as described previously [10]. Cytokines present in supernatants obtained from in vitro culture and in ex vivo single cells isolated from the lungs, spleens and lymph nodes of immunised or Mtb-infected mice were measured using commercial ELISA kits as per the manufacturers’ instructions after stimulation with the PPD, ESAT-6, and HSP90-E6 (2 μg/mL). PPD was kindly provided by Dr. Brennan, at Aeras (Rockville, MD, USA). All ELISA kits were purchased from eBioscience, except for the IL-10 ELISA kit (BD Bioscience).

### 2.8. Cell Culture

Murine bone marrow-derived DCs were generated, cultured, and purified, as described in a previous study [10]. BMDMs were prepared using recombinant M-CSF, as previously described [10]. Briefly, bone marrow cells isolated from C57BL/6 mice were lysed with red blood cell (RBC)-lysing buffer (ammonium chloride 4.15 g/500 mL, 0.01 M Tris-HCl buffer pH 7.5 ± 0.2) and washed with the RPMI 1640 medium. The obtained cells were plated in six-well culture plates (10^6^ cells/mL, 3 mL/well) and cultured at 37 °C in the presence of 5% CO_2_ in RPMI 1640 media supplemented with 100 unit/mL penicillin/streptomycin (Lonza), 10% of fetal bovine serum (Lonza), 50 μM mercaptoethanol (Lonza), 0.1 mM nonessential amino acids (Lonza), 1 mM sodium pyruvate (Sigma), 20 ng/mL GM-CSF, and 10 ng/mL IL-4 (BMDCs) or 20 ng/mL M-CSF (BMDMs).

### 2.9. In Vitro T-Cell Proliferation and Polarisation Assay

CD4^+^ T cells were purified by CD4^+^ T Cell Isolation Kit using a LS column (Miltenyi Biotec) from total mononuclear cells extracted from individually vaccinated C57BL/6J mice. These T-cells were stained with 1 μM CFSE (Invitrogen) as previously described [33]. DCs (2 × 10^5^ cells per well) treated with 2 μg/mL of PPD, E6, or HSP90-E6 for 24 hr were co-cultured with CFSE-stained and CD4^+^ T-cells (2 × 10^6^) at DC:T-cell ratios of 1:10. On day 3 or 4 of co-culture, each T-cell batch was stained with anti CD4^+^, T-bet, or RORγt mAbs, and analysed by flow cytometry.

### 2.10. Colocalisation of Phagosomes and Phagolysosomes

To observe the colocalisation of Mtb-containing phagosomes, we performed confocal microscopy as described previously [34]. Macrophages (2 × 10^5^/well) were prepared in 12-well culture dishes that contained 18 mm diameter round glass coverslips. The cells were then infected with Mtb-RFP at an MOI of 1 for 4 h at 37 °C in a 5% CO2 incubator and incubated with (i) recombinant mIFN-γ (1 ng/mL), recombinant mIL-17 (1 ng/mL), or recombinant mIFN-γ/mIL-17 (1 ng/mL), (ii) re-stimulated with ESAT-6 in vaccinated-lung cells with/without IFN-γR1 (200 ng/mL), IL-17RA (200 ng/mL), or IFN-γR1/IL-17RA (200 ng/mL). After 72 hr incubation, the cells were stained with anti-LAMP1 (Santa Cruz, CA, USA) and imaged under a confocal microscope.

### 2.11. Statistical Analysis

All the experiments were repeated at least three times with consistent results. For immunological analysis, the levels of significance for comparison between samples were determined by Tukey’s multiple comparison or unpaired *t*-test. For CFU and histopathology analysis, Mann-Whitney rank test was used when comparing the differences between two different groups. For statistical analysis, GraphPad Prism version 6.00 for Windows was used (GraphPad Software, La Jolla California USA, www.graphpad.com). The data in the graphs are expressed as the mean ± SD. Differences having * *p* < 0.05, ** *p* < 0.01, *** *p* < 0.001, or **** *p* < 0.0001 were considered statistically significant.

## 3. Results

### 3.1. Characterisation of the Immune Responses induced by HSP90-E6

To examine whether HSP90-E6 induces an antigen (Ag)-specific memory T-cell response, we analysed Ag-specific IFN-γ, TNF-α, IL-2, IL-4, IL-10, and IL-17 production in the lungs, spleen, and lymph nodes of mice 4 weeks after the last immunisation and before challenge (Figure 1a, green arrow). All Ag-specific Th1 cytokines, except IL-4 and IL-10, were significantly induced in mice immunised with ESAT-6 or HSP90-E6 when compared to mice immunised with BCG alone (Figure 1b,c). When stimulated with purified protein derivative (PPD) antigen, IFN-γ, TNF-α and IL-2 production in the lungs, spleen, and lymph nodes of mice immunised with HSP90-E6 was significantly higher than that in mice immunised with BCG or ESAT-6 (Figure 1b). Notably, upon re-stimulation with ESAT-6, not only Th1 cytokines, but also the Th17-related cytokine IL-17 was significantly produced in all three tissues evaluated in the HSP90-E6 immunised mice, when compared to the other treatment groups (Figure 1c). These results suggested that HSP90-E6/CIA05-boosting establishes Th1/Th17-biased immunity.

### 3.2. HSP90-E6/CIA05 induces Ag-Specific Multifunctional T-Cells

Although there is no consensus on correlates of protection against TB in TB vaccine development, recent studies in animal models have suggested the protective contribution of multifunctional T-cells and a Th17-mediated immune response against Mtb infection [13,28,29,35]. Therefore, we next assessed the generation of Ag-specific IL-17-, IFN-γ-, TNF-α- and IL-2-producing multifunctional T-cells upon ex vivo re-stimulation with ESAT-6 after the final immunisation. CD4^+^ T-cells collected from the lungs, spleen, and lymph nodes were stained for intracellular cytokines and subtyped by multi-colour flow cytometry (Appendix A). We focused on multifunctional T-cells producing IL-17 and IFN-γ for comparative analysis. Upon re-stimulation with ESAT-6 or PPD, HSP90-E6/CIA05 immunisation induced expansion of Ag-specific CD4^+^CD44^+^ multifunctional T-cells (IFN-γ^+^IL-17^+^TNF-α^+^IL-2^+^, IFN-γ^+^IL-17^+^IL-2^+^, IFN-γ^+^IL-17^+^TNF-α^+^, and IFN-γ^+^IL-17^+^ cells) in the lungs, spleen, and lymph nodes to a similar extent as ESAT-6/CIA05 immunisation. These multifunctional T-cells were more strongly expanded in HSP90-E6/CIA05- than in ESAT-6/CIA05-immunised mice, especially in the spleen (Figure 2, Appendix A).

### 3.3. Protective Efficacy of HSP90-E6 Boosting against Hypervirulent Mtb HN878

Given the early protective role of IL-17 and the protective efficacy of HSP90-E6/MPL-DDA vaccination against Mtb HN878 [10,28], any Ag that can enhance the limited BCG efficacy against emerging Mtb strains during long-term infection can be regarded a novel vaccine target. In this context, we evaluated the protective efficacy of HSP90-E6/CIA05 suggested by the immunisation-induced Ag-specific multifunctionality, including IFN-γ^+^IL-17^+^ T-cells (Figure 1 and Figure 2). Mice were challenged with Mtb HN878 4 weeks after the last immunisation, and the bacterial burden in and histopathology of the lungs and spleen were evaluated 10 weeks post-infection (Figure 1a). In HSP90-E6/CIA05-boosted mice, lung inflammation, and lesion size were significantly ameliorated when compared to the BCG-vaccinated mice or ESAT-6/CIA05-boosted mice group (Figure 3a,b). Moreover, bacterial burden in the lungs and spleen was significantly lower in HSP90-E6/CIA05-boosted than in ESAT-6/CIA05-boosted mice (Figure 3c). These results suggested that HSP90-E6 vaccination can boost the protection induced by BCG.

### 3.4. Multifunctional T-Cells and Cytokine Profiles in Mice Immunised with BCG Prime HSP90-E6/CIA05 Boosting after Challenge with Mtb HN878

Based on the immunological contribution of vaccination-induced Th1/Th17-biased immunity (Figure 1) and the enhanced protection (Figure 3), we next evaluated whether the magnitude and quality of poly-functionalities upon Ag re-stimulation, represented by IFN-γ^+^ and IL-17^+^, could be sustainably induced or expanded. At 10 weeks after HN878 challenge, lung cells (Figure 4a) and splenocytes (Appendix A) were stimulated ex vivo with PPD or ESAT-6, and CD4^+^ Ag-specific T-cells were typed by multi-colour flow cytometry. In addition, culture supernatants of Ag re-stimulated lung and spleen cells were analysed by enzyme-linked immunosorbent assay (ELISA) (Figure 4b). Robust expansion of Ag-specific CD4^+^CD44^+^ multi-functional (IFN-γ^+^IL-17^+^TNF-α^+^IL-2^+^, IFN-γ^+^IL-17^+^IL-2^+^, IFN-γ^+^IL-17^+^TNF-α^+^ and IFN-γ^+^IL-17^+^) T-cells was observed—especially in lung cells—after stimulation with ESAT-6 or PPD in BCG-primed HSP90-E6/CIA05-boosted, compared to ESAT-6/CIA05-vaccinated mice (Figure 4a). Similar results were obtained in spleen cells stimulated with ESAT-6 (Appendix A). Furthermore, Th1-related IFN-γ, but not Th2-related cytokines, such as IL-4 and IL-10, was markedly increased upon PPD stimulation of lung and spleen cells from fusion protein-boosted mice when compared to the corresponding responses in ESAT-6/CIA05-boosted mice (Figure 4b). Th17-related IL-17 production was higher in lung cells from HSP90-E6/CIA05-boosted than in those from ESAT-6/CIA05-boosted mice (Figure 4b). Upon ESAT-6 stimulation, protective cytokines in lungs and spleen where more strongly induced in HSP90-E6/CIA05- than in ESAT-6/CIA05-boosted mice. Interestingly, IL-2 production was significantly up-regulated in lung cells and splenocytes upon PPD or ESAT-6 re-stimulation, suggesting that increased IL-2 production contributes to the quality of T-cell responses. These results suggested that HSP90-E6 vaccination can boost protective Ag-specific multifunctional T-cells, especially CD4^+^CD44^+^ IFN-γ^+^IL-17^+^ T-cells.

### 3.5. Correlates of Protection for Improved BCG Boosting Efficacy of BCG-HSP90-E6/CIA05 in the Lungs

Next, we aimed to identify immunological correlates for the improved BCG-primed protection conferred by HSP90-E6/CIA05 boosting. Given that the subunit candidate elicited predominantly CD4^+^-biased T-cell responses, positive correlates associated with enhanced protection against highly virulent Mtb infection were analysed based on the experimental set-up in this study. We previously reported that growth inhibition of Mtb in HSP90-E6/MPL-immunised mice is related with the activation of CD4^+^CD44^+^IFN-γ^+^TNF-α^+^IL-2^+^ multifunctional T-cells [10]. In line herewith, these multifunctional T-cells were expanded in the lungs of HSP90-E6/CIA05-immunised mice before (Appendix A) and after (Appendix A) challenge. To identify protective biomarkers for clinical or experimental evaluation of TB vaccine candidates, CD4^+^ T-cell subsets expressing various combinations of four cytokines were compared among the different treatment groups. As shown in Figure 5a, the CD4^+^CD44^+^IFN-γ^+^IL-17^+^ T-cell (R = −0.8641, *p* < 0.0001) responses to ESAT-6 in BCG-HSP90-E6/CIA05-vaccinated mice correlated with bacterial loads in both pre-and post-infections; CD4^+^CD44^+^IFN-γ^+^IL-17^+^TNF-α^+^IL-2^+^ T-cells (R = −0.9082, *p* < 0.0001), CD4^+^CD44^+^IL-17^+^ T-cells (R = −0.8183, *p* < 0.0001), and CD4^+^CD44^+^IL-17^+^IL-2^+^ T-cells (R = −0.6333, *p* = 0.0002) appeared to be significantly correlated only in pre-infected mice, which suggested that these cell populations could be the key factors correlating with BCG-HSP90-E6/CIA05 vaccination-induced protection. On the other hand, CD4^+^CD44^+^IFN-γ^+^IL-17^+^IL-2^+^ T-cells (R = −0.7133, *p* < 0.0001) and CD4^+^CD44^+^IFN-γ^+^TNF-α^+^IL-2^+^ T-cells (R = −0.6845, *p* < 0.0001) showed correlation with bacterial load only post-infection. However, the CD4^+^CD44^+^IFN-γ^+^ T-cell (R = 0.6789, *p* < 0.0001) responses to ESAT-6 correlated negatively with bacterial load post-infection, but not pre-infection (Figure 5b). No prominent correlation was observed in other multifunctional T-cells (Appendix A). The correlation results are summarised in Appendix A by ranking. Taken together, these data suggested that Ag-specific IFN-γ^+^IL-17^+^ responses are involved in HSP90-E6/CIA05-mediated BCG boosting effects.

### 3.6. Th1- and Th17-Related Responses are Simultaneously induced in DCs and Co-Cultured CD4^+^ T-Cells upon Stimulation with HSP90-E6

Recent studies have suggested that IFN-γ/IL-17 responses contribute to protection against Mtb infection [13,28,36,37,38,39]. We previously reported that proinflammatory cytokines such as IFN-γ and IL-17, which are related to anti-mycobacterial activity, were significantly induced by T-cells activated by HSP90-matured DCs [10]. Therefore, we next tested whether HSP90-E6-matured DCs produce cytokines related to Th1/Th17 polarisation. Interestingly, the fusion protein significantly stimulated the secretion of TNF-α, IL-1β, IL-12p70, and IL-23p19, but not IL-10, which have important roles in Th1/Th17 differentiation, in DCs, whereas untreated DCs secreted negligible amounts of these cytokines (Figure 6a). Next, we determined whether HSP90-E6-matured DCs affect cell proliferation. To evaluate the effect of HSP90-E6 on the interaction between DCs and T-cells, we performed in vitro T-cell proliferation assays of carboxyfluorescein succinimidyl ester (CFSE)-labelled sorted CD4^+^ T-cells obtained from vaccinated mice upon co-culture with DCs stimulated with individual vaccine antigens for 72 h. T-cells from mice injected with BCG alone were cocultured with PPD-treated DCs. HSP90-E6-, PPD-, or ESAT-6-treated DCs induced T-cell proliferation primed by immunisation with each Ag to a significantly greater extent than did untreated DCs (Figure 6b). T-cells activated with PPD, ESAT-6, or HSP90-E6-treated DCs produced significantly higher levels of cytokines than those activated with untreated DCs (Figure 6c). The production of IFN-γ, IL-17, and TNF-α was significantly higher in T-cells co-cultured with HSP90-E6-treated DCs than in T-cells co-cultured with E6- or PPD-treated DCs. IL-2 production was similarly induced upon co-culture with all Ag-treated DCs. However, T-cells activated with E6- or HSP90-E6-matured DCs did not significantly produce Th2-related cytokines IL-4 and IL-10, whereas T-cells activated with PPD did (Figure 6c). We next evaluated the differentiation of CD4^+^ T-cells using intracellular transcription factor immunostaining. As shown in Figure 6d, the expression of the Th1/Th17-associated transcription factors T-bet and RoRγt was increased in proliferating T-cells co-cultured with HSP90-E6-treated DCs, as compared to the levels in T-cells co-cultured with E6-treated DCs (Figure 6d). Taken together, these results suggested that HSP90-E6 is capable of simultaneously polarizing T-cells towards Th1 and Th17.

### 3.7. IFN-γ and IL-17 Synergistically Exert Anti-Mycobacterial Activity Via Enhanced Phagolysosomal Maturation

The involvement of IL-17 in the protection against Mtb infection has been controversial, as the interaction between Th1 and Th17 cells is not well understood [13,30,40,41]. Th17 responses reportedly do not inhibit the generation of Th1 cells in vitro, and favour Th1 responses in vivo [41]. Thus, we hypothesised that IL-17 alone does not effectively inhibit Mtb growth within macrophages, but may act synergistically with IFN-γ. To investigate this hypothesis, Mtb-infected macrophages were treated with IFN-γ, IL-17 or both. IL-17 alone did not inhibit intracellular Mtb growth, even at concentrations up to 100 ng/mL, whereas IFN-γ alone significantly inhibited Mtb growth at 1 ng/mL, with no significant difference between 1 ng/mL and 10 ng/mL of IFN-γ (Appendix A). Interestingly, IFN-γ plus IL-17 at 1 ng/mL each significantly inhibited Mtb growth in macrophages, when compared to 1 ng/mL IFN-γ (Figure 7a), and IFN-γ plus IL-17 at 5 ng/mL each significantly inhibited Mtb growth, when compared to 100 ng/mL IFN-γ (Appendix A). Mtb growth was inhibited in a concentration-dependent manner upon treatment with both cytokines (Appendix A). These results suggested that IL-17 synergistically enhances the anti-mycobacterial activity of IFN-γ.

We previously demonstrated that T-cells activated with HSP90-E6-matured DCs inhibit Mtb growth in macrophages [10]. Therefore, we further investigated the synergistic effect of both cytokines on HSP90-E6-mediated anti-mycobacterial activity after inhibition of cytokine signalling by using neutralizing anti-cytokine antibodies. As expected, Mtb-infected macrophages co-cultured with T-cells activated by HSP90-E6-matured DCs significantly inhibited intracellular bacterial growth, when compared to infected macrophages co-cultured with T-cells activated by untreated DCs (Figure 7b). Pre-treatment with anti-IFN-γ or both anti-IFN-γ and anti-IL-17 completely abrogated HSP90-E6-mediated Mtb growth inhibition via T-cell activation. There was no difference in inhibitory activity between anti-IFN-γ and both anti-IFN-γ and anti-IL-17. The anti-IL-17 antibody alone also showed inhibitory activity, but less than the anti-IFN-γ antibody, indicating that IL-17 synergistically enhances the antimycobacterial activity of IFN-γ. Next, we evaluated the roles of both cytokines in Mtb-infected macrophages exposed to supernatants from ESAT-6-re-stimulated lung cells of HSP90-E6-vaccinated mice before and after challenge. Ag-stimulated culture supernatants significantly inhibited Mtb growth in macrophages when compared to untreated culture supernatants (Figure 7C), and these inhibitory effects were abrogated upon addition of anti-IFN-γ and/or anti-IL-17. Supernatants from ESAT-6-re-stimulated spleen cells of HSP90-E6-vaccinated mice before and after challenge showed a similar Mtb growth-inhibitory effect, which was also neutralised by anti-IFN-γ and/or anti-IL-17 (Appendix A). However, supernatants from ESAT-6-re-stimulated spleen and lung cells collected from BCG-vaccinated naïve mice before Mtb challenge did not inhibit Mtb growth in macrophages, and supernatants from ESAT-6-re-stimulated spleen and lung cells of E6-boosted mice showed a limited inhibitory effect (Appendix A). Culture supernatants from the spleen and lung cells of BCG-vaccinated mice after Mtb challenge did not inhibit Mtb growth in macrophages and culture supernatants from the lung, but not in the spleen cells of E6-boosted mice, which showed a limited inhibitory effect (Appendix A). These results suggested that effector T-cells with antimycobacterial activity are effectively expanded in HSP90-E6-boosted mice.

Next, we examined whether IFN-γ and IL-17 could affect phagosome-lysosome fusion, which is the Mtb-killing mechanism of macrophages. As shown in Figure 8a, IL-17 alone did not induce phagosome maturation, but it did significantly enhance IFN-γ-mediated colocalisation of Mtb with the lysosomal marker LAMP1. Mtb–LAMP1 colocalisation was more strongly induced upon co-treatment with IFN-γ and IL-17 (1 ng/mL each) than after treatment with IFN-γ (1 ng/mL) alone (Figure 8a), and the level of colocalisation was comparable to that after treatment with 100 ng/mL IFN-γ (Appendix A). IFN-γ and IL-17 did not show a synergistic effect on IFN-γR1 expression in Mtb-infected macrophages (Appendix A), nor on reactive oxygen species (ROS) production, which was increased by IFN-γ or IL-17 treatment (Appendix A). As expected, IFN-γ, but not IL-17 induced nitric oxide (NO) production in Mtb-infected macrophages, and IFN-γ-mediated NO production was decreased upon IL-17 treatment (Appendix A). Supernatants from ESAT-6-re-stimulated lung cells of HSP90-E6-vaccinated mice significantly enhanced Mtb–LAMP1 colocalisation, which was decreased upon treatment with anti-IFN-γ or anti-IL-17 (Figure 8b). These data suggested that antigens capable of simultaneously inducing Th1 and Th17 responses may have strong TB vaccine potential.

## 4. Discussion

The present study revealed that the TB vaccine candidate HSP90-E6, given as a BCG-prime boost regimen, confers superior, long-term protection against hypervirulent Mtb HN878 infection when compared to BCG or BCG-E6 alone. Further, elevated E6-specific CD4^+^ IFN-γ^+^IL-17^+^ T-cells pre- and post-infection positively correlated with protection against Mtb.

There clearly is an urgent, unmet need for new anti-TB vaccines, and heterologous prime-boost vaccination appears to be a promising strategy. However, despite increased research efforts on this approach in the last two decades, clinical progress has been limited, due to insufficient understandings on protective immune response, targeting antigens, and scheme of boosting. A number of fusion protein-based subunit vaccines are being tested as boosters to BCG. We previously demonstrated that HSP90-E6 formulated with MPL/DDA significantly reduced the bacterial load in mouse lungs after challenge with HN878 [10], but we did not identify an immunologic correlate of protection for this fusion vaccine. Here, we report that HSP90-E6 formulated with CIA05/DDA prolongs BCG-primed boosting and that its protective effect is related to enhancement of Ag-specific-IFN-γ^+^IL-17^+^ multifunctional T-cells.

The primary rationale for the development of TB vaccines designed to elicit Th1-cell-based immunity is based on evidence from various animal models that a strong IFN-γ-mediated Th1 immune response is the primary protective mechanism of anti-TB immunity [19,35,42,43,44,45]. However, an IFN-γ response is not an optimal correlate of protection [16,17,18], and an IFN-γ response alone is not sufficient to control Mtb infection [46]. Our data suggest that the CD4^+^ T-cell populations that mediate immunity to TB are likely not solely mediated by IFN-γ-producing Th1 cells, but also by cells whose effector function is independent of IFN-γ [47,48]. A previous study demonstrated that antigens that do not elicit Th1 responses uniformly fail to protect against Mtb, but not all proteins that induce robust Th1 responses after vaccination provide considerable protection [49]. In addition, Th17 cells reportedly also contribute to protective TB immunity in mice [28,29,50], cynomolgus macaques [51], and rhesus macaques [52]. Although the role of Th17 cells in human patients appears ambiguous [41,53,54], early Th17 responses are suppressed in progressors as compared to Mtb-infected, healthy controls [55]. Therefore, we analysed Ag-specific T-cell responses to identify immunological correlates for the superior BCG-primed HSP90-E6 booster vaccination. Notably, BCG-primed HSP90-E6 booster vaccination simultaneously elicited Th1/Th17-biased immune responses after the last immunisation, especially upon ESAT-6 re-stimulation of the lung and spleen cells, when compared to BCG-primed ESAT-6-boosted vaccination (Figure 1). In addition, BCG boost with HSP90-E6/CIA05 induced a significant increase in the number of Ag-specific CD4^+^CD44^+^IL-17^+^ T-cells co-producing three or two effector cytokines after in vitro stimulation with ESAT-6 in the spleen cells (Figure 2), but not the lymph-node and lung cells, which was expected, considering that adjuvanted antigens were injected intramuscularly. We infer that ready-to-be-expanded IFN-γ/IL-17-producing multifunctional CD4^+^ T-cells exist, and that their Ag-specific expansion induced by HSP90-E6 boosting possibly contributes to the enhanced protection against Mtb infection.

Based on these results, we evaluated the protective effect of BCG-prime HSP90-E6/CIA05 vaccination against the hypervirulent HN878 strain in a mouse model. HSP90-E6/CIA05 vaccination markedly enhanced the protective efficacy of BCG at 10 weeks post-infection, as indicated by reduced bacterial loads and smaller inflamed lesions (Figure 3). Furthermore, BCG-prime HSP90-E6 vaccination induced an exclusive quadruple cytokine-positive population, expressing both IL-17 and IFN-γ in addition to IL-2 and TNF-α. The ESAT-6-specific IL-17-, IFN-γ-, TNF-α-, and IL-2-producing CD4^+^ T-cell population in the lungs and spleen was considerably larger than that in BCG-immunised and BCG-primed ESAT-6-boosted mice, and Ag-specific CD4^+^CD44^+^ T-cells in the lungs were expanded when compared to pre-infection numbers (Figure 4). Pre-infection expansion of IFN-γ/IL-17-producing cells in the lungs was inversely correlated with bacterial burden (R = −0.8641, *p* < 0.0001), but there was no significant correlation between IFN-γ-producing cell responses and lung bacterial burden (R = −0.0679, *p* = 0.726). Notably, the expansion of Ag-specific CD4^+^IFN-γ^+^IL-17^+^ T-cells was identified as a correlate of protection after Mtb challenge (R = −0.7703, *p* < 0.0001), indicating that these cells are an exclusive feature of BCG-primed HSP90-E6 vaccination. Given that Ag-specific IFN-γ responses were similarly induced in all groups of immunised mice, an IFN-γ response alone is insufficient to control Mtb infection (R = 0.6789, *p* < 0.0001), especially in the absence of IL-17 contribution (Figure 5).

It was difficult to detect differences between the groups when multifunctional T-cell responses were presented as percentages of the parental populations; therefore, we used actual cell counts to evaluate T-cell responsiveness against Ag and how many Ag-responding T-cells existed in the Mtb-infected tissues. Upon vaccination but prior to infection, CD4^+^CD44^+^ T-cell numbers generally increased in all organs, when compared to the numbers in naïve mice. Additionally, splenic CD4^+^CD44^+^ T-cells displayed increased infiltration of cells capable of producing cytokines in response to Ag stimulation in both boosted groups, which may be because of the route of Ag immunisation. Further, multifunctional T-cells were more highly expanded in HSP90-E6-than in ESAT-6-boosted mice (Figure 2). Upon infection, the expansion of PPD-/ESAT-6-specific multifunctional T-cells in the lungs and ESAT-6-specific multifunctional T-cells in the spleen were more obvious in HSP90-E6- than in ESAT-6-boosted mice, which is linked to protective efficacy of HSP90-E6, which outperforms BCG and BCG-prime ESAT-6 vaccination (Figure 3 and Figure 4, and Appendix A).

In vitro, HSP90-E6 also significantly induced DCs possessing a Th1/Th17-polarizing phenotype, characterised by IL-1β, IL-12p70, and IL-23p19 production, as compared to ESAT-6 alone. Moreover, IFN-γ and IL-17 production in T-cells from individually vaccinated mice was markedly increased upon co-culture with HSP90-E6- versus ESAT-6-treated DCs, and was accompanied with enhanced T-cell proliferation (Figure 6), demonstrating that HSP90-E6 is capable of inducing Th1/Th17-biased responses mediated by optimal DC activation. Most importantly, IFN-γ and IL-17 clearly synergistically contributed to intracellular bacterial growth inhibition in Mtb-infected macrophages through enhanced phagolysosomal fusion (Figure 7 and Figure 8), whereas ROS, NO, and up-regulation of IFN-γR1 were not involved in the synergistic effect (Appendix A). Thus, we identified an unappreciated role for IFN-γ/IL-17 synergism in inducing anti-bacterial phagosomal activity for the control of Mtb infection in the host, in addition to the existing paradigm that IL-17 rapidly promotes cell recruitment, and thereby contributes to early granuloma formation in infection sites.

In summary, our study revealed that HSP90-E6/CIA05 vaccine exerts a durable BCG-boosting effect against hypervirulent HN878 Mtb in mice via expansion of IFN-γ/IL-17-producing cells in the lungs. The number of IFN-γ-producing T-cells in the lungs was significantly increased in only infection group, but decreased in HSP90-E6/CIA05-immunised group. This suggests that IFN-γ-producing T-cells are necessary, but not sufficient for TB defence, and, more importantly, that an increase in the number of cells that produce both IFN-γ and IL-17 is important for protection. Further, our results indicate that the presence of both IFN-γ and IL-17 in the lungs upon Mtb challenge may be an essential immunological signature of balanced inflammation and protection. The immune signatures associated with superior BCG-primed HSP90-E6 booster vaccination provide important leads for further investigation of the direct role of IL-17 in protection against TB and a novel strategy to improve BCG-booster vaccines. Thus, IFN-γ/IL-17-producing multifunctional CD4^+^ T-cells are determinants of protective efficacy of TB subunit vaccines, and HSP90-E6/CIA05 is an excellent TB vaccine candidate that effectively induces such a response.

## 5. Conclusions

Collectively, out study demonstrated that IFN-γ-producing T-cells are necessary, but not sufficient for TB defense, and, more importantly, that an increased in the number of cells that produce both IFN-γ and IL-17 is essential for protection. This protective immune determinant related with BCG-primed HSP90-E6 booster vaccination will pave the way for further investigation of a novel strategy to improve BCG-booster vaccines.

## Figures and Tables

**Figure 1 vaccines-08-00300-f001:**
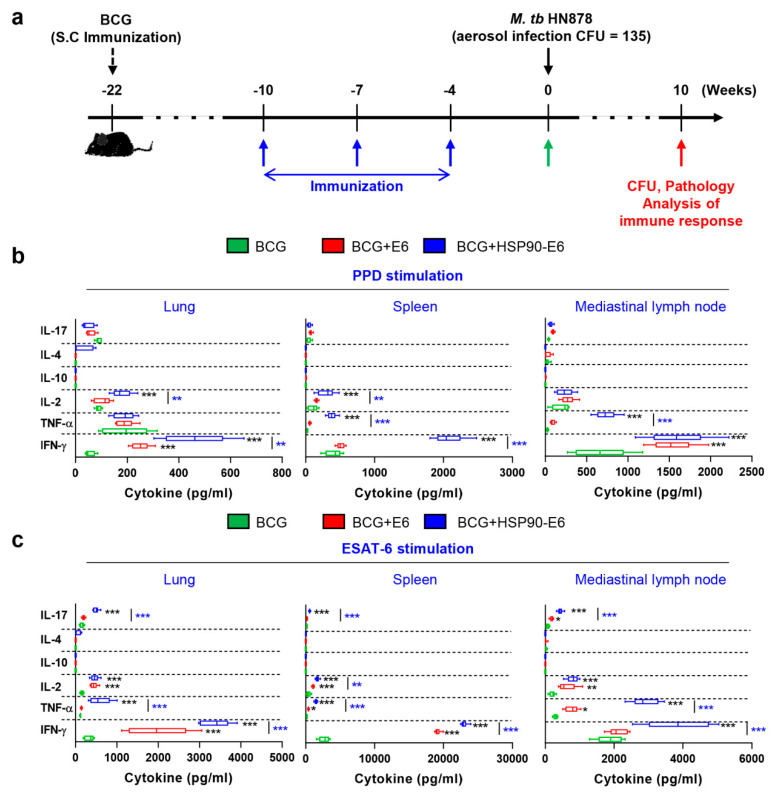
Comparative cytokine production profiles in Bacillus Calmette-Guérin (BCG)-, BCG+ESAT-6-, or BCG+HSP90-E6-vaccinated mice upon antigen (Ag) stimulation four weeks after final vaccination. (**a**) Experimental design for HSP90-E6 subunit vaccine testing. Mice (n = 12 per group) were immunised by BCG injection 12 weeks before subunit vaccination. Three intramuscular injection of HSP90-E6/CIA05 were administered (blue arrows) before *Mycobacterium tuberculosis* (Mtb) HN878 aerosol challenge (black arrow). Immunological analysis was conducted before (green arrow) and after Mtb infection (red arrow). Bacterial counts and histopathology in each group were determined at the indicated time points after Mtb infection (red arrow). (**b**,**c**) Levels of IFN-γ, TNF-α, IL-2, IL-10, IL-4, and IL-17 secreted by lung, spleen, and lymph-node cells from each fully immunised group in response to ESAT-6 (2 μg/mL) or purified protein derivative (PPD) (2 μg/mL) stimulation as detected by enzyme-linked immunosorbent assay (ELISA). * *p* < 0.05, ** *p* < 0.01 and *** *p* < 0.001 compared to BCG-immunised mice. ** *p* <0.01 and *** *p* < 0.001 between BCG+ESAT-6- and BCG+HSP90-E6-immunised mice.

**Figure 2 vaccines-08-00300-f002:**
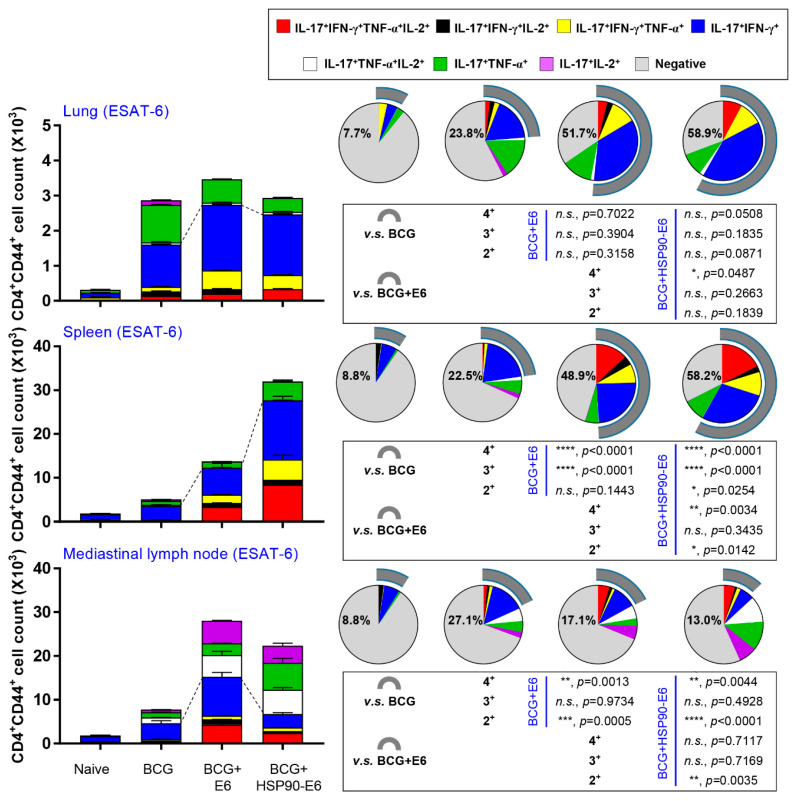
Induction of Ag-specific multifunctional T-cells in the lungs, spleen, and lymph nodes in BCG+HSP90-E6-immunised mice. Mice were immunised and euthanised, as described in the Methods section. Four weeks after the last immunisation, the mice were sacrificed, and their lung, spleen, and lymph-node cells collected from the mice were treated with ESAT-6 (2 μg/mL) at 37 °C for 12 h in the presence of GolgiStop. Upon stimulation with ESAT-6, cell counts of Ag-specific, multifunctional CD4^+^CD44^+^ T-cells producing IFN-γ, IL-17 and/or TNF-α and IL-2 in the lung, spleen, and lymph-node cells from each immunised group were determined by flow cytometry. Gray arc denotes the percentage of cytokine-positive T-cells (IL-17^+^IFN-γ^+^TNF-α^+^IL-2^+^-, IL-17^+^IFN-γ^+^IL-2^+^-, IL-17^+^IFN-γ^+^TNF-α^+^-, and IL-17^+^IFN-γ^+^-CD4^+^CD44^+^ T-cells). The figure 2+ stands for sum percentages of double-cytokine positive T-cells (IL-17^+^IFN-γ^+^, IL-17^+^TNF-α^+^, and IL-17^+^IL-2^+^), 3+ stands for triple-cytokine positive T-cells (IL-17^+^IFN-γ^+^IL-2^+^, IL-17^+^IFN-γ^+^TNF-α^+^ and IL-17^+^TNF-α^+^IL-2^+^), and 4+ stands for quadruple-cytokine positive T-cells (IL-17^+^IFN-γ^+^TNF-α^+^IL-2^+^). Data are expressed as the mean ± SD for five mice from each group. *n.s.*: not significant, * *p* < 0.05, ** *p* < 0.01, *** *p* < 0.001 and **** *p* < 0.0001 compared to BCG-immunised mice. *n.s.*: not significant, * *p* < 0.05, ** *p* < 0.01 between BCG+ESAT-6- and BCG+HSP90-E6-immunised mice.

**Figure 3 vaccines-08-00300-f003:**
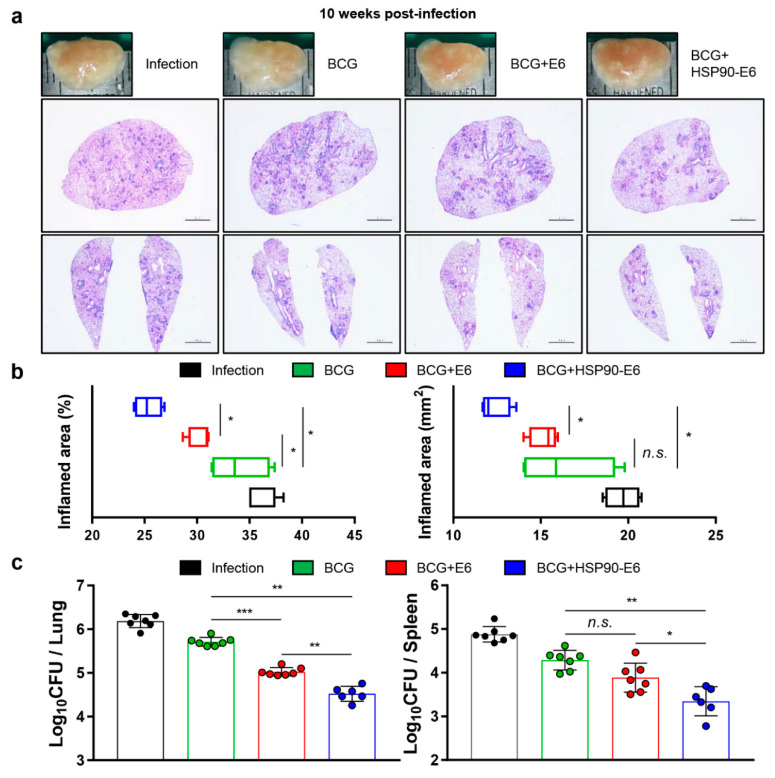
HSP90-E6/CIA05 booster vaccination improves BCG-primed protection against hypervirulent Mtb HN878. (**a**) H&E staining of superior lobes of the right lung of each immunised mouse at 10 weeks after Mtb HN878 infection (scale bars = 2.0 mm). (**b**) Inflamed area as % of the total area (left) and lesion size (right) in the lungs. (**c**) Colony forming units (CFUs) in the lungs and spleen in all treatment groups at 10 weeks post-infection, determined by counting the viable bacteria. Data are from one of two independent experiments (*n* = 6 or 7 mice per group at each time point). Mann-Whitney rank tests were used to compare groups. *n.s.*: not significant, * *p* < 0.05, ** *p* < 0.01, and *** *p* < 0.001.

**Figure 4 vaccines-08-00300-f004:**
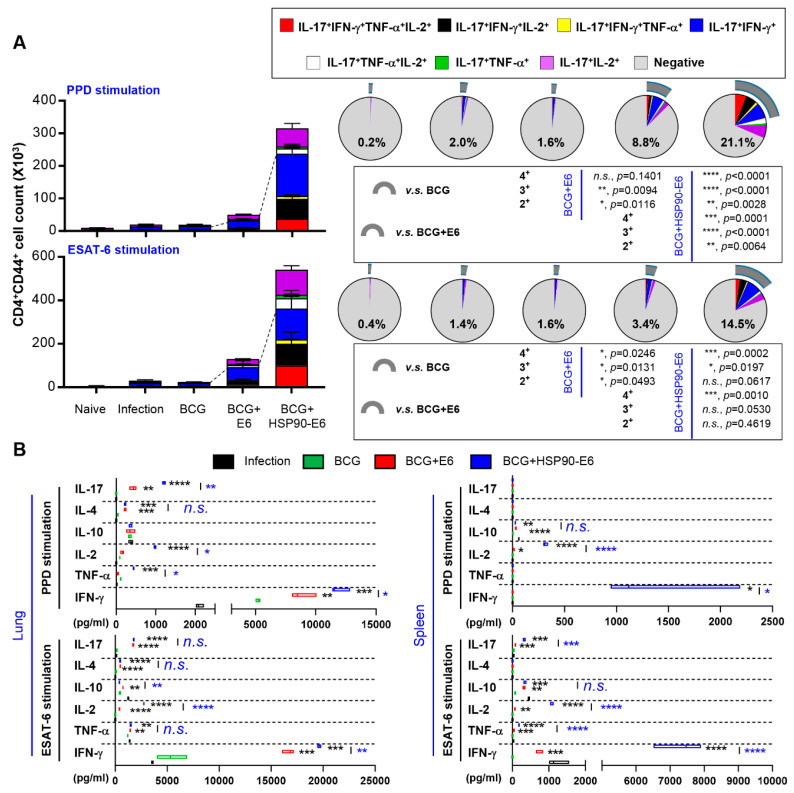
Ag-specific multifunctional T-cell subsets and cytokine production after challenge with Mtb HN878. (**A**) Mice of each group were sacrificed 10 weeks post-infection, and spleen and lung cells obtained from the mice were treated with ESAT-6 (2 μg/mL) at 37 °C for 12 h in the presence of GolgiStop. Upon stimulation with the ESAT-6, cell counts of Ag-specific, multifunctional CD4^+^CD44^+^ T-cells producing IFN-γ, IL-17 and/or TNF-α and IL-2 in the spleen and lung cells in all treatment groups were determined by flow cytometry. Gray arc denotes the percentage of cytokine-positive T-cells (IL-17^+^IFN-γ^+^TNF-α^+^IL-2^+^-, IL-17^+^IFN-γ^+^IL-2^+^-, IL-17^+^IFN-γ^+^TNF-α^+^-, and IL-17^+^IFN-γ^+^-CD4^+^CD44^+^ T-cells). The figure 2+ stands for sum percentages of double-cytokine positive T-cells (IL-17^+^IFN-γ^+^, IL-17^+^TNF-α^+^, and IL-17^+^IL-2^+^), 3+ stands for triple-cytokine positive T-cells (IL-17^+^IFN-γ^+^IL-2^+^, IL-17^+^IFN-γ^+^TNF-α^+^ and IL-17^+^TNF-α^+^IL-2^+^), and 4+ stands for quadruple-cytokine positive T-cells (IL-17^+^IFN-γ^+^TNF-α^+^IL-2^+^). Data are the mean ± SD for 6 or 7 mice from each group. (**B**) Levels of IFN-γ, TNF-α, IL-2, IL-10, IL-4, and IL-17 secreted by lung and spleen cells in all treatment groups in response to ESAT-6 (2 μg/mL) or PPD (2 μg/mL) stimulation as detected by ELISA. *n.s.*: not significant, * *p* < 0.05, ** *p* < 0.01, *** *p* < 0.001 and **** *p* < 0.0001 compared to BCG-immunised mice. *n.s.*: not significant, ** *p* < 0.01, *** *p* < 0.001 and **** *p* < 0.0001 between BCG+ESAT-6- and BCG+HSP90-E6-immunised mice.

**Figure 5 vaccines-08-00300-f005:**
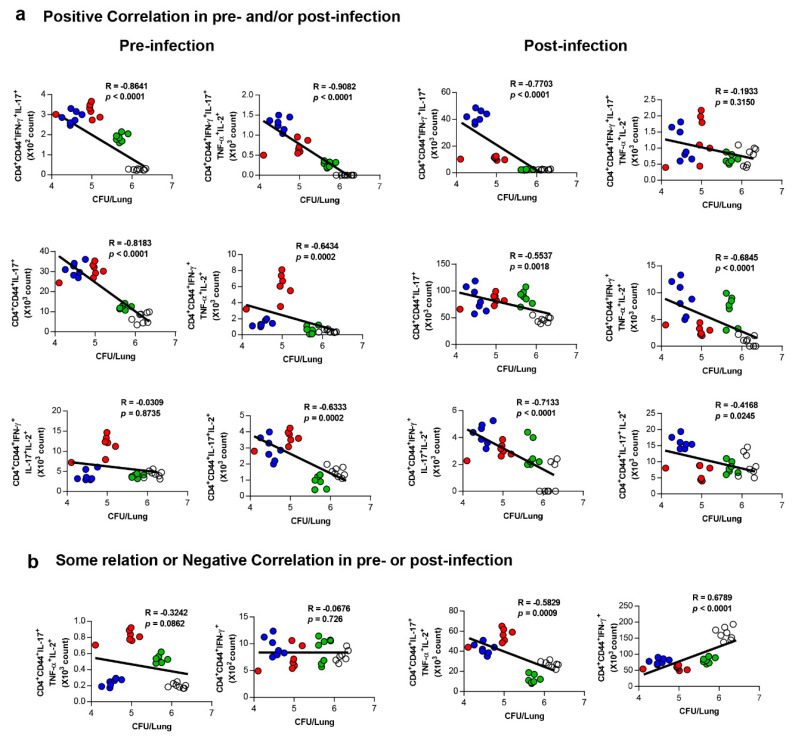
Analysis of protective correlations for protection level with vaccine-induced immune responses pre- and post-infection. Relationship between protection (CFU) and ESAT-6 specific CD4^+^CD44^+^IFN-γ^+^, CD4^+^CD44^+^IFN-γ^+^IL-17^+^TNF-α^+^IL-2^+^, CD4^+^CD44^+^IFN-γ^+^IL-17^+^IL-2^+^, CD4^+^CD44^+^ TNF-α^+^ producing T-cells or CD4^+^CD44^+^IFN-γ^+^IL-17^+^ is shown as a fitted regression line with the correlation coefficient. (**a**) Positive correlation, (**b**) some relation or negative correlation. Spearman’s r and P values of the correlations are indicated. White circle: naïve or infection, green circle: BCG, red circle: BCG+E6, and blue circle: BCG+HSP90-E6.

**Figure 6 vaccines-08-00300-f006:**
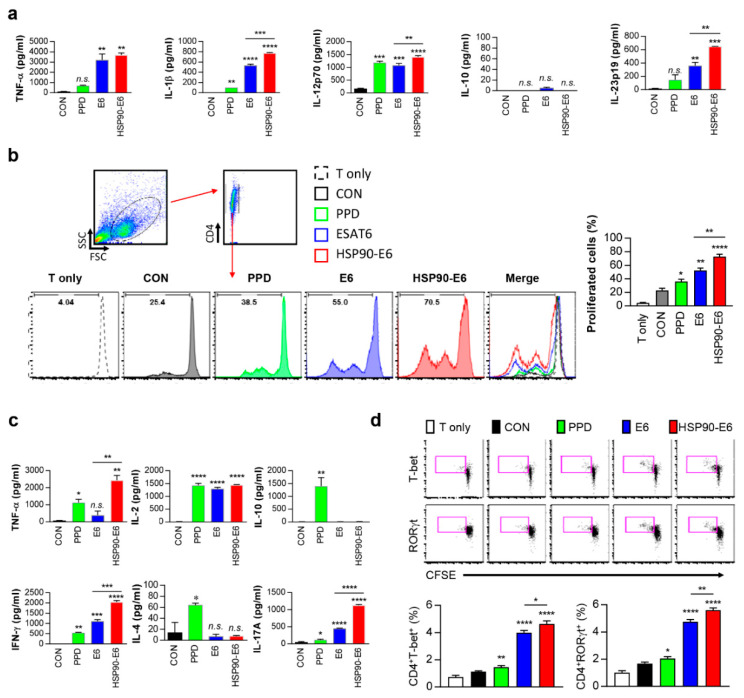
T-cell proliferation and differentiation induced by HSP90-E6-treated dendritic cells (DCs). (**a**) DCs were activated with PPD (2 μg/mL), ESAT-6 (2 μg/mL), or HSP90-E6 (2 μg/mL) for 24 h. TNF-α, IL-1β, IL-12p70, IL-10, and IL-23p19 levels in the culture medium were measured by ELISA. Data are the mean ± SD (n = 3). (**b**) CD4^+^ T-cells were isolated from the spleens of individually vaccinated mice, stained with carboxyfluorescein succinimidyl ester (CFSE), and co-cultured with DCs treated with PPD (2 μg/mL), E6 (2 μg/mL), or HSP90-E6 (2 μg/mL) for 3 days. T-cells alone and T-cells co-cultured with untreated DCs served as controls. CD4^+^ T-cell proliferation was assessed by flow cytometry. (**c**) Culture supernatants were harvested after 3 days and TNF-α, IFN-γ, IL-2, IL-4, IL-10, and IL-17A secretion levels were measured by ELISA. (**d**) Expression of Th1- and Th17-related transcription factors was assessed using intracellular staining after 4 days of co-culture with DC:T-cells (CON: Naïve T-cells, PPD: BCG vaccinated T-cells, E6: ESAT-6/CIA05 immunised T cells, HSP90-E6: HSP90-ESAT-6/CIA05 immunised T cells). Data are the mean ± SD from three independent experiments; * *p* < 0.05 versus appropriate controls. *n.s.*: no significant difference, * *p* < 0.05, ** *p* < 0.01, *** *p* < 0.001 and **** *p* < 0.0001 compared to untreated DCs. * *p* < 0.05, ** *p* < 0.01, *** *p* < 0.001 and **** *p* < 0.0001 between ESAT-6- and HSP90-E6-treated DCs.

**Figure 7 vaccines-08-00300-f007:**
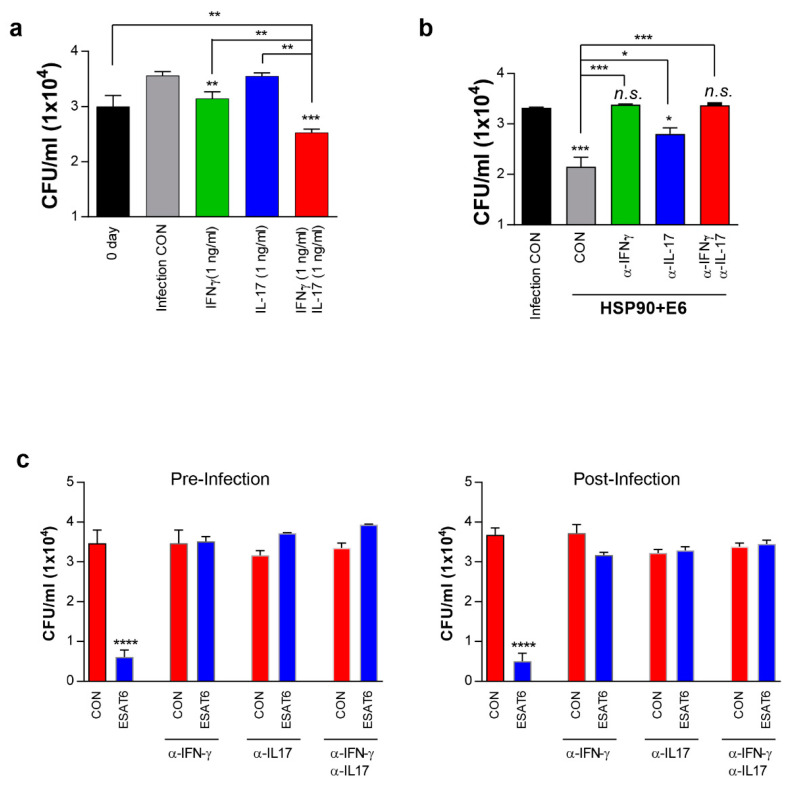
Synergistic inhibition of intracellular Mtb growth in macrophages by Th1/Th17-induced responses. (**a**) Mtb-infected bone marrow-derived macrophages (BMDMs) were treated with IFN-γ (1 ng/mL), IL-17 (1 ng/mL), or both (1 ng/mL each) for 3 days. Intracellular Mtb growth in the BMDMs was determined at time point 0 days and 3 days after cytokine treatment. Data are the mean ± SD (*n* = 3); * *p* < 0.05, ** *p* < 0.01, or *** *p* < 0.001. *n.s.*: no significant difference. (**b**) T-cells activated with unstimulated DCs or HSP90-E6-stimulated DCs at a DC:T-cell ratio of 1:10 for 3 days were co-cultured with Mtb-infected BMDMs in the presence or absence of neutralising antibody (anti-IFN-γ or anti-IL-17). Intracellular Mtb growth in BMDMs was determined after 3 days of co-culture, with or without T-cells (control). Data are the mean ± SEM (n = 3). (**c**) BMDMs were treated with supernatants of ESAT-6-re-stimulated HSP90-E6-vaccinated lung cells in the presence or absence of anti-IFN-γ or anti-IL-17 for 3 days. Intracellular Mtb growth in the BMDMs was determined after 3 days. Data are the mean ± SD (*n* = 3); * *p* < 0.05, ** *p* < 0.01, *** *p* < 0.001 and **** *p* < 0.0001. *n.s.*: no significant difference.

**Figure 8 vaccines-08-00300-f008:**
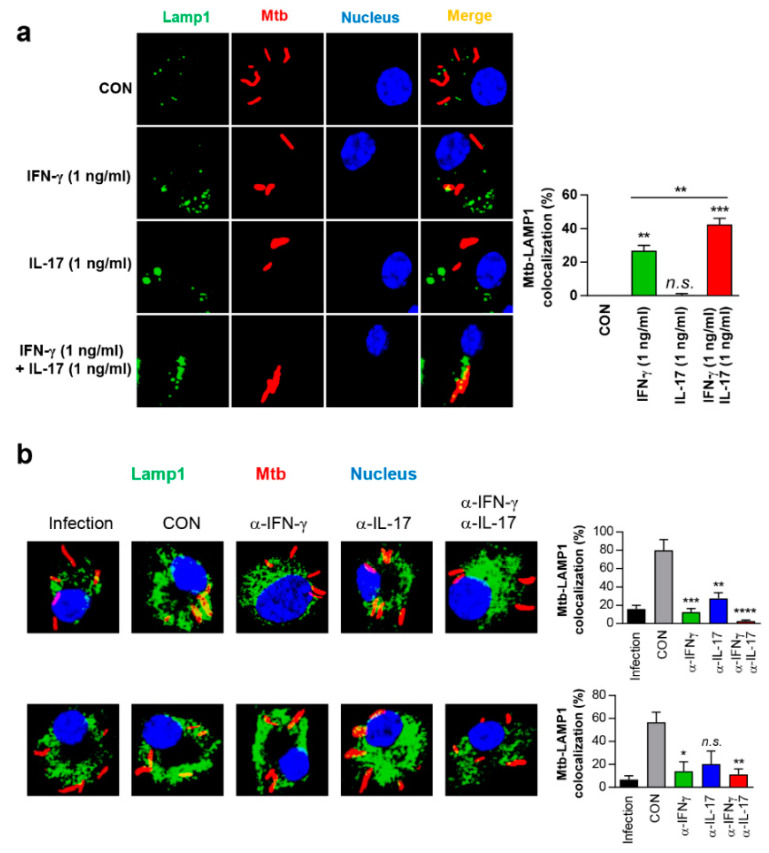
IFN-γ and IL-17 synergistically inhibit intracellular bacterial growth via phagosome-lysosome fusion. BMDMs were infected with Mtb-RFP (MOI = 1) for 4 h, washed, incubated with/without IFN-γ, IL-17, or IFN-γ/IL-17 (1 ng/mL) (**a**) or with supernatants of ESAT-6-re-stimulated in HSP90-E6-vaccinated lung cells with/without anti-IFN-γ or anti-IL-17 (**b**) for 72 h, fixed with 4% paraformaldehyde, and immunolabeled with anti-LAMP1 antibody and Alexa 488-conjugated goat anti-rabbit or anti-rat IgG (green). Nuclei were counterstained with DAPI (chromosome counterstain, 4′,6-diamidino-2-phenylindole, blue). The cells were analysed by laser-scanning confocal microscopy. Scale bar, 10 μm. Quantification of Mtb–LAMP1 colocalisation is shown in the bar graph. Data are the mean ± SD of 50–100 cells per experiment (*n* = 3). * *p* < 0.05, ** *p* < 0.01, *** *p* < 0.001 and **** *p* < 0.0001 versus infection only control (CON) or for differences between treatments. *n.s*., no significant difference.

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
