# Peer review of "Antigen-Specific IFN-γ/IL-17-Co-Producing CD4+ T-Cells are the Determinants for Protective Efficacy of Tuberculosis Subunit Vaccine"

_vaccines, 2020, doi:10.3390/vaccines8020300_

Round 1

Reviewer 1 Report

I would like to congratulate the authors on a thorough evaluation of cytokine synergy in MTB innate immune responses, as well as correlates of adaptive immune protection. This study therefore makes an important contribution to the understanding of pathogen clearance mechanisms, and can be used to help improve vaccine design for TB. The quality of the results and report mean that I could only find very minimal changes to suggest. I especially like the inclusion of total cell count numbers when comparing groups, as I think this a very pertinent biological readout when evaluating vaccine efficacy. Also, reducing the bacterial CFU load by several logs is quite impressive in TB challenge models. Once again, well done on a nice study and best of luck for the future development of your vaccine candidates.

Specific recommendations that could be included in a revised manuscript:

  1. Could the authors please include a brief statement or explanation of why heterologous prime boost vaccines might be particularly effective/desirable/advantageous, or mention their advantages over single dose or homologous prime boost? From a manufacture/clinical development/regulatory approval standpoint, heterologous pharmaceuticals seem like they would have additional complication or restrictions during vaccine development.

2. To show specificity of your main FACS readout (cytokine positive cells), please show a stimulated vs unstimulated control sample in supplementary Figure 1 so that your placement of cytokine gates can be validated.

Minor editing:

The greek symbols, e.g.the  gamma symbol of IFN-g and the micron symbol, do not appear correctly in most of the manuscript, eg. line 28 on page 2. and line 11 on page 4

Please specify in the methods section the exact type of MACS kit used for cell purification, and explain the method in slightly more detail.

Future recommendations:

Many researchers in the TB vaccine field are quantifying/characterising the cellular response in distinct parts of the lung (upper airways vs progression into the lower lung, or alternatively BAL vs vascular vs interstitial (CD103+ CD69+ CXCR3+) tissue trafficking/residence of T cells because there is increasing evidence that cell location or trafficking potential is a key factor in . I therefore recommend you perform intravenous fluorescent CD45 staining one or two minutes before sacrificing the mice to be able to distinguish whether the cells you isolate from the lung are truly in the lung parenchyma  or BAL (CD45-) or in the lung bloodstream (CD45+), which is where they would be poised to exert more rapid responses upon airway challenge.

Author Response

[Reviewer 1]

Comments and Suggestions for Authors

I would like to congratulate the authors on a thorough evaluation of cytokine synergy in MTB innate immune responses, as well as correlates of adaptive immune protection. This study therefore makes an important contribution to the understanding of pathogen clearance mechanisms, and can be used to help improve vaccine design for TB. The quality of the results and report mean that I could only find very minimal changes to suggest. I especially like the inclusion of total cell count numbers when comparing groups, as I think this a very pertinent biological readout when evaluating vaccine efficacy. Also, reducing the bacterial CFU load by several logs is quite impressive in TB challenge models. Once again, well done on a nice study and best of luck for the future development of your vaccine candidates.

Specific recommendations that could be included in a revised manuscript:

[Q1] Could the authors please include a brief statement or explanation of why heterologous prime boost vaccines might be particularly effective/desirable/advantageous, or mention their advantages over single dose or homologous prime boost? From a manufacture/clinical development/regulatory approval standpoint, heterologous pharmaceuticals seem like they would have additional complication or restrictions during vaccine development.

[A1] We feel very thankful to the reviewer for your deep and thorough comments. Although homologous prime-boost immunization is very effective at generating antibody responses, it is generally inefficient for boosting cellular immunity, which is a key element in protection against intracellular pathogens. In contrast, heterologous prime-boost strategies have been successfully used to enhance cellular immunity against Mtb. The initial priming results as elicitation of primary antigen-specific T-cell responses, some of which subsequently become antigen-specific memory T cells. When a boosting agent is given, these memory T cells expand, forming a larger pool of immune memory (PMID: 27269058, 29847865, 15196215, Book: Malaria: Immune Response to Infection and Vaccination). One of the advantages of using protein vaccine is safety and long term memory effect. We appreciate again for suggesting your highly interesting future work for single dose related experiments.

[Q2] To show specificity of your main FACS readout (cytokine positive cells), please show a stimulated vs unstimulated control sample in supplementary Figure 1 so that your placement of cytokine gates can be validated.

[A2] We thank the referee for the excellent comments and suggestions, which were extremely valuable for improving the quality of our manuscript. To validate the cytokine-producing T cell subsets, gates of positive staining for each cytokine were delineated based on unstimulated controls. As suggested, we have added the gating strategy used to gate the cytokine-producing T cells from unstimulated control [Supplementary Figure 1].

Minor editing:

[Q3] The greek symbols, e.g.the gamma symbol of IFN-g and the micron symbol, do not appear correctly in most of the manuscript, eg. line 28 on page 2. and line 11 on page 4

[A3] There may be something wrong during PDF conversion when submitted. We apologize for our mistake. We have checked out the overall manuscript once again and have corrected it.

[Q4] Please specify in the methods section the exact type of MACS kit used for cell purification, and explain the method in slightly more detail.

[A4] As your comments, we have added the information about MACS kit in the Material and Method section [Page 5, Line 19]. Also, we rewrote the method with more details.

Future recommendations:

Many researchers in the TB vaccine field are quantifying/characterising the cellular response in distinct parts of the lung (upper airways vs progression into the lower lung, or alternatively BAL vs vascular vs interstitial (CD103+ CD69+ CXCR3+) tissue trafficking/residence of T cells because there is increasing evidence that cell location or trafficking potential is a key factor in . I therefore recommend you perform intravenous fluorescent CD45 staining one or two minutes before sacrificing the mice to be able to distinguish whether the cells you isolate from the lung are truly in the lung parenchyma or BAL (CD45-) or in the lung bloodstream (CD45+), which is where they would be poised to exert more rapid responses upon airway challenge.

[A] We would like to thank the reviewer for the excellent recommendations. We will track whether the isolated cells actually exist in the lung parenchyma and BAL (CD45-), or the lung bloodstream (CD45+) through the experiment method you suggested in further experiments.

Reviewer 2 Report

Revision of manuscript vaccines-811657

Dear Authors,

Your manuscript entitled “Antigen-specific IFN-γ/IL-17-co-producing CD4+ T- cells are the determinants for protective efficacy of tuberculosis subunit vaccine” reports an interesting evaluation of a potential candidate as tuberculosis vaccine. The work was well planned and conducted. Results were clearly presented and appropriately discussed in all aspects. The work is of sure great interest and usefulness; furthermore, it opens to new other investigations.

Probably the manuscript could be accepted in present form, but below you Authors can find some suggestions:

  • The “Greek letters” are missing all over the manuscript (as: “IFN-, TNF-,” or “with PPD (2 g/ml) or ESAT-6 (2 g/ml)”); probably some errors occur during paper formatting.
  • In my opinion, it would be better to explain protocols involving mice, like for example: number of animals for groups; method for euthanasia; time of animals killing for the various protocols (like for “3.1. Characterisation of the immune responses induced by HSP90-E6”); some of these informations are reported in Results section and in figure’s heading, but they are not well presented in Material and Method section.

I sincerely hope that these suggestions will enhance this manuscript. However, if I have made any errors or misinterpretations, I apologize in advance.

Sincerely

The Reviewer

Author Response

 [Reviewer 2]

Dear Authors,

Your manuscript entitled “Antigen-specific IFN-γ/IL-17-co-producing CD4+ T- cells are the determinants for protective efficacy of tuberculosis subunit vaccine” reports an interesting evaluation of a potential candidate as tuberculosis vaccine. The work was well planned and conducted. Results were clearly presented and appropriately discussed in all aspects. The work is of sure great interest and usefulness; furthermore, it opens to new other investigations.

Probably the manuscript could be accepted in present form, but below you Authors can find some suggestions:

[Q1] The “Greek letters” are missing all over the manuscript (as: “IFN-, TNF-,” or “with PPD (2 g/ml) or ESAT-6 (2 g/ml)”); probably some errors occur during paper formatting.

[A1] We apologize for our mistake. We checked out the overall manuscript once again.

[Q2] In my opinion, it would be better to explain protocols involving mice, like for example: number of animals for groups; method for euthanasia; time of animals killing for the various protocols (like for “3.1. Characterisation of the immune responses induced by HSP90-E6”); some of these informations are reported in Results section and in figure’s heading, but they are not well presented in Material and Method section.

[A2] We admit our mistake for missing an important point. We have added the information about animals in each group [Page 3, Line 26], method for euthanasia, time of animals killing [Page 3, Line 39] in the Material and Method section. Also, we rewrote the method with more details.

Reviewer 3 Report

By using the TB vaccine candidate, HSP90-E6, the authors identified a novel role of multifunctional Th cells producing IFNg and IL-17 in protection against Mtb.
There were two important findings in the paper. First, BCG followed by HSP90-E6 vaccination induced multifunctional antigen-specific Th cells producing IFNg and IL-17. Second, IFNg and IL-17 synergistically inhibited bacterial growth in macrophages.
They proposed a story in which, HSP90-E6 vaccination stimulates DCs to promote Th cell differentiation into multifunctional Th cells. Then, these Th cells secrete cytokines IFNg and IL-17, which control macrophages by enhancing bacteriocidal activity and inhibit bacterial growth. This story is very interesting and clearly described.
This story may describe the mechanism of the efficacy of HSP90-E6 as TB vaccine. It also suggests that Th17-inducing protocols may be useful for TB vaccine. Therefore, this paper is very informative and worth publishing.

While I read the manuscript, I got some questions as listed below. These points should be clarified before publication.

Materials and Methods
Preparation of antigens (PPD, E6, HSP90-E6) should be described.

About antigens
For in vitro stimulation of T cells, why two antigens (PPD and ESAT)-6 were used? These two antigens led to almost same results, but there some differences (i.e. IL-17 production in Fig. 1b and c). Can the author explain the difference?

page 4, line 8-10
BMDCs were incubated with the mixture of DCs and T cells. It was better to incubate with culture supernatants of DC-T cell culture?

page 4, line 45
pH7.5 ± 2, Is it true? Or pH7.5 ± 0.2?

page 5, line 3
How mice for T cell preparation were vaccinated (BCG, BCG+E6, or BCG+HSP90-E6)?

page 5, line 4
How DCs were activated? Were they incubated with purified proteins without any DC-stimulating reagents? If so, did only ESAT-6 stimulate DCs to secrete cytokines? (Fig. 6a)

page 8, Fig. 2 (and Fig.4a)
Information in the legend is not enough so that I cannot understand how to read these plots.
Do the bar plots and pie charts express same data?
What represent percentages in pie charts?
What are gray arches beside the pie charts?
What do “2+, 3+, 4+” mean?
line 9, mean ± SD. But there is no error bar.

page 11, Fig. 4b
The scale should be shown (pg/mL?)

page 12, Fig. 5
It is not clear whether “CD4+ CD44+ IFNg+ IL-17+” means “CD4+ CD44+ IFNg+ IL-17+ TNFa+and− IL-2+and−” or “CD4+ CD44+ IFNg+ IL-17+ TNFa− IL-2−”.

page 12, Fig. 5a
What do #1 and #2 mean?

page 13, line 12-14
The sentence “We previously reported that ... HSP90-maturated DC [10].” may be wrong.

page 14, Fig. 6c
As DCs secreted TNFa (Fig. 6a), it is difficult to conclude that T cells secreted TNFa. DCs without T cells may be included as a control.

page 17, Fig. 8b
What is CON? Only supernatant, or control Ab?

Supplemental information
page 2
The plots of isotype control should be shown, otherwise, the validity of gating cannot be evaluated

Throughout the text
problems in characters in Symbol font, e.g. gannma in IFN-g, alpha in TNF-a, micro in microgram, gamma in RORgT.

Author Response

[Reviewer 3]

Comments and Suggestions for Authors

By using the TB vaccine candidate, HSP90-E6, the authors identified a novel role of multifunctional Th cells producing IFNg and IL-17 in protection against Mtb.

There were two important findings in the paper. First, BCG followed by HSP90-E6 vaccination induced multifunctional antigen-specific Th cells producing IFNg and IL-17. Second, IFNg and IL-17 synergistically inhibited bacterial growth in macrophages.

They proposed a story in which, HSP90-E6 vaccination stimulates DCs to promote Th cell differentiation into multifunctional Th cells. Then, these Th cells secrete cytokines IFNg and IL-17, which control macrophages by enhancing bactericidal activity and inhibit bacterial growth. This story is very interesting and clearly described.

This story may describe the mechanism of the efficacy of HSP90-E6 as TB vaccine. It also suggests that Th17-inducing protocols may be useful for TB vaccine. Therefore, this paper is very informative and worth publishing.

While I read the manuscript, I got some questions as listed below. These points should be clarified before publication.

Materials and Methods

[Q1] Preparation of antigens (PPD, E6, HSP90-E6) should be described.

[A1] We appreciate these excellent comments. As suggested, we have added the information of antigen preparation in Materials and Methods section based on our previously published paper (PMID: 28193909) in the revised manuscript.

About antigens

[Q2] For in vitro stimulation of T cells, why two antigens (PPD and ESAT)-6 were used? These two antigens led to almost same results, but there some differences (i.e. IL-17 production in Fig. 1b and c). Can the author explain the difference?

[A2] We appreciated this valuable comment. We intended to use PPD and ESAT-6 antigens for ex vivo stimulation of T cells. Firstly, we evaluated the immunogenicity whether both BCG immunisation was well-established with stimulation of PPD. By the usage of ESAT-6 which is a shared antigen between ESAT-6- and HSP90-E6-immunised mice, we were able to check whether immunogenicity has been well-achieved by subunit immunisation. Secondly, PPD and ESAT-6 induced the differential IL-17 production profiles as you mentioned. Upon ESAT-6 re-stimulation, IL-17 production was only displayed in HSP90-E6-immunised mice compared to E6-immunised mice. Although it is not described in this paper, weak IFN-g responses were induced with the stimulation of HSP90 in HSP90-E6-immunised mice. Collectively, such differences are likely to be resulted from antigen-specificity.

[Q3] BMDCs were incubated with the mixture of DCs and T cells. It was better to incubate with culture supernatants of DC-T cell culture? (page 4, line 8-10)

[A3] We apologize for the confusion. We clearly described that the mixture was replaced with culture supernatants in the revised manuscript. [Page 4, Line 25]

[Q4] pH7.5 ± 2, Is it true? Or pH7.5 ± 0.2? (page 4, line 45)

[A4] Thank you for your indication. We have corrected it in the revised manuscript. [Page 5, Line 11]

[Q5] How mice for T cell preparation were vaccinated (BCG, BCG+E6, or BCG+HSP90-E6)? (page 5, line 3)

[A5] We apologize for not explaining this information in detail. Mice were immunized as described in Materials and Methods section, then DCs stimulated with each antigen (PPD, ESAT-6, or HSP90-E6) were co-cultured with T cells from individually vaccinated mice (BCG, BCG+E6, or BCG+HSP90-E6) respectively. We have added detailed information in the legend section of Figure 6 [Page15, Line 10-11].

[Q6] How DCs were activated? Were they incubated with purified proteins without any DC-stimulating reagents? If so, did only ESAT-6 stimulate DCs to secrete cytokines? (Fig. 6a) (page 5, line 4)

[A6] You are correct. DCs were only activated by Mtb antigens.

[Q7] Information in the legend is not enough so that I cannot understand how to read these plots.

Do the bar plots and pie charts express same data?

What represent percentages in pie charts?

What are gray arches beside the pie charts?

What do “2+, 3+, 4+” mean?

line 9, mean ± SD. But there is no error bar. (page 8, Fig. 2 (and Fig.4a))

[A7] We apologize for the confusion. The bar graphs and pie charts expressed the same data in different way based on numbers and percentages respectively. In addition, the percentages in pie charts stand for proportion of IL-17+IFN-g+TNF-a+IL-2+-, IL-17+IFN-g+IL-2+-, IL-17+IFN-g+TNF-a+-, and IL-17+IFN-g+-CD4+CD44+ T cells among total CD4+CD44+ T cells, and gray arc simply emphasizes the percentage of cytokine-positive T cells. Moreover, 2+ stands for sum percentages of double-cytokine positive T cells (IL-17+IFN-g+, IL-17+TNF-a+, and IL-17+IL-2+), 3+ stands for triple-cytokine positive T cells (IL-17+IFN-g+IL-2+, IL-17+IFN-g+TNF-a+ and IL-17+TNF-a+IL-2+), and 4+ stands for quadruple-cytokine positive T cells (IL-17+IFN-g+TNF-a+IL-2+) for evaluating quality of T cell responses. Lastly, we have added the error bars in bar graphs in the revised manuscript [Figure 2 and Figure 4a].

[Q9] The scale should be shown (pg/mL?) (page 11, Fig. 4b)

[A9] We apologize for missing the information. As suggested, we have added this information in the revised manuscript [Figure 4b].

[Q10] It is not clear whether “CD4+ CD44+ IFNg+ IL-17+” means “CD4+ CD44+ IFNg+ IL-17+ TNFa+and− IL-2+and−” or “CD4+ CD44+ IFNg+ IL-17+ TNFa− IL-2−”. (page 12, Fig. 5)

[A10] We apologize for the confusion. CD4+ CD44+ IFNg+ IL-17+ means CD4+ CD44+ IFNg+ IL-17+ TNFa− IL-2−. The other population of the correlation information was described in the supplementary table 1.

[Q11] What do #1 and #2 mean? (page 12, Fig. 5a)

[A11] We apologize for the confusion. We have deleted unnecessary expressions from Figure 5a.

[Q12] The sentence “We previously reported that ... HSP90-maturated DC [10].” may be wrong. (page 13, line 12-14)

[A12] We apologize for the confusion. We have replaced the sentence for clarification. [Page 13, Line 13-15]

[Q13] As DCs secreted TNFa (Fig. 6a), it is difficult to conclude that T cells secreted TNFa. DCs without T cells may be included as a control. (page 14, Fig. 6c)

[A13] We appreciate your comments. We intended to perform two independent experiments. First experiment (Figure 6a) was about evaluating DC activation after each antigen stimulation, while second experiment (Figure 6b-d) was performed for evaluating the capacity of T cell activation driven by antigen-stimulated DCs. Importantly, antigen-stimulated DCs were washed with PBS before the co-culture. We believe that this process may prevent TNF-a secretion from DCs. We apologize for the confusion.

[Q14] What is CON? Only supernatant, or control Ab? (page 17, Fig. 8b)

[A14] We apologize for the confusion. CON was only supernatants of ESAT-6-re-stimulated in HSP90-E6-vaccinated lung cells

Supplemental information

[Q15] The plots of isotype control should be shown, otherwise, the validity of gating cannot be evaluated (page 2)

[A15] We thank the referee for the excellent comments and suggestions, which were extremely valuable for improving the quality of our manuscript. To validate the cytokine-producing T cell subsets, gates of positive staining for each cytokine were delineated based on unstimulated controls. As suggested, we have added the gating strategy used to gate the cytokine-producing T cells from unstimulated control [Supplementary Figure 1].

Throughout the text

[Q16] problems in characters in Symbol font, e.g. gannma in IFN-g, alpha in TNF-a, micro in microgram, gamma in RORgT.

[A16] We apologize for our mistake. We checked out the overall manuscript once again.